# Stem Cells in Regenerative Medicine: A Journey from Adult Stem Cells to Induced Pluripotent Cells

**DOI:** 10.3390/ijms26178255

**Published:** 2025-08-26

**Authors:** Ylenia Della Rocca, Antonella Mazzone, Guya Diletta Marconi, Oriana Trubiani, Jacopo Pizzicannella, Francesca Diomede

**Affiliations:** 1Department of Innovative Technologies in Medicine & Dentistry, “G. d’Annunzio” University of Chieti-Pescara, Via dei Vestini, 31, 66100 Chieti, Italy; ylenia.dellarocca@unich.it (Y.D.R.); antonella.mazzone@unich.it (A.M.); guya.marconi@unich.it (G.D.M.); oriana.trubiani@unich.it (O.T.); francesca.diomede@unich.it (F.D.); 2Department of Engineering and Geology, “G. d’Annunzio” University of Chieti-Pescara, Viale Pindaro, 42, 65127 Pescara, Italy

**Keywords:** human adult stem cells, iPSCs, human embryonic development, molecular biology, regenerative medicine

## Abstract

Regenerative medicine is the branch of medicine that aims to repair and regenerate damaged tissues and presents promising avenues for addressing a wide range of currently incurable diseases. Regenerative medicine is based on the use of cell therapy with stem cells that can differentiate into differentiated cells of specific tissues. There are various types of stem cells, which are different in potential and derivation. The aim of this review is to summarize the types of stem cells most studied and recently discovered, from adult stem cells to innovative induced pluripotent stem cells (iPSCs), for regenerative medicine purposes. The stem cells involved in the identification of new regenerative therapeutic approaches are analyzed here through a classification based on the tissues’ embryonic derivation: stem cells from ectodermal derivation tissues, stem cells from mesodermal derivation tissues, stem cells from endodermal derivation tissues, and iPSCs.

## 1. Introduction

Regenerative medicine offers solutions for many incurable diseases [1]. Regenerative medicine is a multidisciplinary field of health sciences that aims to repair, regenerate, or replace dysfunctional cells, tissues, or organs through different methods and applications [2]. Currently, end-stage organ disease is treated with approaches like the support of organ function through dialysis, respiratory devices, or ventricular pumps, and eventually transplantation, which are not always curative. The endogenous regeneration or supply of de novo generated cells and tissues, thanks to regenerative techniques, is an alternative to transplantation [3]. The basis of regenerative medicine is cell therapy with somatic, adult stem, or embryo-derived cells [4]. Stem cells have one of the greatest impacts on the field of regenerative therapies. Stem cells are cells capable of self-renewal and differentiation into specialized tissue cells, depending on the stem cells’ potency, whether they are totipotent, pluripotent, multipotent, or unipotent [5]. Multipotent stem cells, such as hematopoietic stem cells (HSCs), can develop into multiple specialized cells in a specific tissue. Induced pluripotent stem cells (iPSCs), like embryonic stem cells (ESCs), are pluripotent stem cells capable of differentiating into cells of all three embryonic lineages [6].

Regenerative medicine offers cellular therapeutic perspectives for various diseases.

In the context of liver diseases, different cell therapies have been evaluated in clinical settings, like inborn errors of metabolism, acute liver failure, chronic liver disease, liver cirrhosis, and acute-on-chronic liver failure, with different cell sources, such as hepatocytes, liver progenitor cells, biliary tract stem/progenitor cells, mesenchymal stromal cells, and macrophages, through different administration routes [7]. In gastrointestinal tract diseases, intestinal stem cell (ISC) transplantation allows for damaged intestinal mucosa to be repaired and opens up new regenerative therapies for the treatment of inflammatory bowel diseases (IBDs) [8]. Stem-cell-based regenerative therapies provide approaches that also modulate neurogenesis. Some diseases, such as Alzheimer’s, affect neurogenesis; these specific conditions that stimulate endogenous neurogenesis may be helped with the promotion of the regenerative and recovery process in this disease [9]. Stem cell trials provide new treatments for a variety of neurological diseases by potentially restoring impaired neurological function due to disease, maldevelopment, or trauma. Although several stem cell trials have shown clinical benefit in diseases like demyelinating diseases/spinal cord injury, amyotrophic lateral sclerosis, stroke, Parkinsons, Huntington’s, macular degeneration, and peripheral nerve diseases, there has not always been a cell replacement mechanism but rather a local “neuroprotective” effect, trophic support, and influence on the immune system, providing indirect benefits to the nervous system [10]. In the orthopedic and sports medicine field, regenerative medicine can help to deal with various musculoskeletal conditions in treatments called ortho-biological, which are proving to be innovative in terms of their effect and potential efficacy [11].

Osteoarthritis (OA) is one of the diseases studied as treatable with regenerative medicine approaches that can stop or slow its progression, especially if treated early before irreversible joint destruction occurs [12]. Stem cells and regenerative medicine hold great promise for the treatment of chronic lung diseases through the development of new disease models for the study of lung pathophysiology and for the identification of personalized therapies [13]. Regenerative medicine approaches have also been used to treat diseased tissues and organs of the reproductive system in experiments on both animal models and humans, showing promising results for the treatment of reproductive system inadequacies [14]. Various types of regenerative medicine methods, including cell therapy with autologous epidermal melanocyte/keratinocyte cells or mesenchymal stem cells, have been reviewed for application in specific dermatological disorders, reporting improvement in androgenetic alopecia and vitiligo. Other diseases reviewed were alopecia areata, melasma, lichen sclerosus et atrophicus (LSA), inflammatory acne vulgaris, chronic telogen effluvium, erosive oral lichen planus, and dystrophic epidermolysis bullosa, for which regenerative medicine has been shown to be an effective treatment option [15]. Cardiovascular diseases (CVDs), the leading cause of death in many countries, can be treated with regenerative medicine techniques to restore cardiac and vascular functions thanks to the ability of stem cells to differentiate into cardiac cells, cardiomyocytes, vascular endothelial cells, and smooth muscle cells and to secrete specific factors that support endogenous repair. Therefore, stem cells, especially autologous stem cells, and factor-based therapies may represent emerging therapeutic options for the treatment of CVD [16]. The aim of this review is to analyze the different types of stem cells, from adult stem cells to iPSCs, as a source for the characterization of new cell-based therapies in regenerative medicine.

## 2. Adult Human Stem Cells

Adult stem cells are undifferentiated cells that are present in all tissues of the body. Adult stem cells can be found in various adult tissues and organs such as bone marrow, oral cavity, brain, skin, and liver, which range from multipotent to unipotent cells and lead to a progeny based on the tissue in which they reside [17]. Adult stem cells can be differentiated into adipocytes, chondrocytes, myocytes, cardiomyocytes, smooth muscle cells, neuronal progenitor cells, and osteoblasts [18]. Adult stem cells are characterized by a high proliferative potential and the ability to differentiate into various cell types, depending on the tissue of origin. Normally, they are in a quiescent state, but they can proliferate and differentiate to replace tissue cells damaged following pathologies and injuries or for the generation of new tissue in regular maintenance [19]. There are several clinical trials for the use of adult stem cells in regenerative medicine, but only a few of them have led to approved therapies. Adult stem cells that are being considered for regenerative medicine approaches are reviewed below, distinguished on the basis of embryonic layer derivation (ectoderm, mesoderm, and endoderm).

### 2.1. Ectoderm-Derived Tissues

During gastrulation (third week of embryonic development), the blastula is single-layered and differentiates into the multilayered gastrula consisting of three distinct layers: ectoderm, mesoderm, and endoderm. The ectoderm is one of the three layers of the early tri-laminar embryo formed. After gastrulation, the process of neurulation begins in mesodermal cells (middle layer of gastrulation), which form the notochord and induce the overlying ectodermal cells in the neural plate to form the neural tube, while the joining ends form the neural crest. The neural tube and neural crest are then separated from the overlying ectoderm. The neural tube develops into the central nervous system, while the neural crest forms other systems, including the peripheral nervous system, the enteric nervous system, and teeth. The remaining ectoderm forms the epidermis of the skin, hair, and exocrine glands [20,21] (Figure 1).

#### 2.1.1. Neural Stem Cells: Neurogenesis and Regenerative Medicine

The central nervous system (CNS) is composed of several cell types, including neurons, astrocytes, oligodendrocytes, and other non-neuronal cells. The main functional cells of the CNS are neurons, which process information entering and leaving the CNS, while astrocytes mainly have the role of trophic support for the CNS and also perform immune-type functions, including phagocytosis. Oligodendrocytes form myelin around axons, allowing the rapid propagation of action potentials. All three of these cell types arise as radial glial cells (RGCs), which then give rise to neuronal progenitors [22]. During embryogenesis, two crucial proliferative zones can be identified: the ventricular zone (VZ) and the subventricular zone (SVZ), which are the primary locations of origin for cortical neurons and glial cells during development [23]. Initially from the neural plate, neuroepithelial cells (NECs) divide symmetrically to maintain their pool and are identified as the earliest form of embryonic neuronal stem cells (NSCs). After the neural tube forms, NECs convert into radial glial cells, whose soma is in the VZ, and the radial fiber runs from the inner surface of the neural tube to the outer surface. Glial cells thus characterized exhibit the properties of embryonic NSCs. During this stage, from each asymmetric division, radial glial cells self-renew on one side and generate a neuronal progenitor on the other [24]. In the late stage of embryogenesis, radial glial cells proliferate to differentiate into oligodendrocytes and finally astrocytes. At the end of pregnancy, toward birth, radial glial cells change characteristics to generate NSCs that serve as a pool of stem cells to support the processes of adult neurogenesis [25]. Different biomarkers have been reported to identify the different cell lineages during neuronal development. Proliferating progenitors residing in the neural tube are characterized by the expression of SRY-Box Transcription Factor 1 (Sox1), which is expressed exclusively in the CNS and probably is the early marker for the differentiation of embryonic stem cells toward the neural phenotype [26]. In the mouse cortex, empty spiracles homolog 2 (Emx2) has been shown to be a transcription factor that participates in the development of proliferating neuroblasts originating from the neuroepithelium, VZ, and postmitotic Cajal–Retzius cells, and is used as a dorsal marker during corticogenesis [27]. Furthermore, Emx2 is involved in the regulation of neuroblast proliferation, migration and differentiation [28]. POU class 3 homeobox 2, also called Brn-2 (Pou3f2), and POU class 3 homeobox 3, also called Brn-1 (Pou3f3), are members of the class III POU family transcription factors involved in neural differentiation [29]. Paired box 6 (Pax6) plays a critical role in NSC proliferation and neuronal fate determination. Pax6 identifies SVZ progenitor cells and has been shown in mice to control neural progenitor cell proliferation by modifying SRY-Box Transcription Factor 2 (Sox2) expression [30]. SRY-Box Transcription Factor 5 (Sox5) may manipulate the migration of previously born neurons from the deep layer to the more superficial layers. In progenitor neuronal cells that reside in VZ and SVZ, Sox5 is not detectable [31]. Brain-like T-box factor 1 (Tbr1) is a transcription factor that cooperates with Sox5 to regulate early-born neurons during embryonic development in the formation of multiple lineages and neuronal migration. Knock-out of Tbr1 in mice has indicated its role in cortical development [32,33]. Tbr1 is not present in progenitor cells residing in the VZ and SVZ, confirming a similar role to Sox5 in function [34]. Fezf2 belongs to the FEZ zinc finger 2 family, which functions as a transcription factor and is involved in the development of the cortical spinal tract. Fezf2 is found in early VZ progenitors and their neuronal progenies but disappears in late progenitor cells and upper layer neurons [33,34]. Cut-like homeobox 1 and 2 (Cux1 and Cux2) participate in late neuronal differentiation, dendritic branching and synaptogenesis in neurons of the upper cortex [35,36]. Although most neurogenesis ends around birth, in limited regions of the adult mammalian brain, neural stem cells (NSCs) continually generate new neurons. Adult neurogenesis appears to be restricted in two regions: V-SVZ of the lateral ventricles, which generates neurons of the olfactory bulb, and the subgranular zone (SGZ) in the hippocampal formation [37]. In adult humans, NSCs are present throughout life in both V-SVZ and SGZ as specialized astrocytes [38]. During adult neurogenesis, NSCs (type 1 cells), are similar to radial glia, although they have distinct morphology [39]. They express the astrocytic marker glial fibrillary acidic protein (GFAP) and nestin [40] and divide asymmetrically to form intermediate progenitors (IPs) (type 2 cells), which express stem/progenitor markers, including Sox2 [41]. At the final stage, the neuronal course becomes evident with the expression of transcription factors including Prospero homeobox protein 1 (Prox1), Neuronal Differentiation 1 (NeuroD1), and doublecortin (DCX) [42]. These cells produce migratory neuroblasts (classified as type 3 cells) that display the polysialylated form of neural cell adhesion molecule (PSA-NCAM) and exhibit DCX-positive and nestin-negative characteristics [39,43]. Subsequently, after proliferation, they start early neuronal development, forming immature neurons that can be marked by DCX, NeuN, and the Ca^2+^-binding protein calretinin [44]. Finally, new mature granule cells expressing calbindin are formed [45]. Neurogenesis in the SGZ is associated with learning and memory but also with mood regulation. Alterations in neurogenesis can cause various pathologies and affective disorders [46]. Furthermore, both V-SVZ and SGZ show a decrease in neurogenesis with aging [47]. Endogenous adult NSCs reside within neurogenic niches and could potentially be exploited for therapeutic and regenerative purposes. Several studies demonstrate that adult NSCs persist into adulthood in humans and can be isolated from brain tissue samples and cultured in vitro, providing insights into the mechanisms of neurogenesis underlying their regenerative potential [48]. For transplantation experiments or clinical applications, it is necessary to purify the NSC population, for example, by fluorescence-activated cell sorting or with magnetic beads. CD71 could be used to distinguish more mature neuronal derivatives compared to early neural progenitor cells, while early-committed NSCs express CD133 and CD15. A2B5, CD44, and CD140a are markers of glial, astrocytic, and oligodendrocytic precursors, respectively. Differentiated neuronal cells express CD56 and CD24 [49]. The regenerative potential of neuroblasts has been evaluated in several studies, mainly on animal models. After stroke, the two events that contribute to the damage repair are neurogenesis and angiogenesis, processes supported by NSCs. In one study, it was demonstrated that transplantation of human embryonic neural stem cells injected into the SVZ of rat models that had undergone induced focal cerebral ischemia increased endogenous cell proliferation in the SVZ and promoted angiogenesis in the peri-infarct area [50]. In murine models of ischemic stroke, newly formed neuroblasts migrate to the sites of ischemic injury and generate synaptic connections with neighboring neurons [51]. In mice with multiple sclerosis (MS), there is an increased influx of V-SVZ-derived cells into sclerotic lesions sites, where they differentiate into oligodendrocytes after demyelination [52]. NSC transplantation is a promising strategy for treating many CNS diseases, also because NSCs express chemokine receptors like CXCR4, and adhesion molecules that mediate migration into damaged tissue can also cause NSCs to attract each other, leading them to cluster [53]. Furthermore, other studies show that NSCs exhibit an anti-inflammatory effect. Intravenous transplantation of NSCs in animals affected by hemorrhagic stroke reduced the cerebral and splenic expression of the inflammatory markers TNF-α, IL-6, and NF-κB, also reducing the microglial infiltrate at the lesion site [54]. More recent approaches aim to genetically modify NSCs to guide the differentiation of transplanted cells and thus enhance therapeutic effects. In a rat model of spinal cord injury, transplantation of NSCs overexpressing Wnt4 shifted their differentiation toward a more neural phenotype. This resulted in better lesion repair and greater functional integration compared to transplantation of unmodified NSCs [55]. Another therapeutic prospect for neurodegenerative diseases is to genetically modify NSCs to express the neurotransmitters that are defective. Park et al. modified a human NSCs line to overexpress the Ach-producing enzyme, choline acetyltransferase (ChAT), to address the cholinergic deficiency that occurs with aging. Compared to unmodified NSCs, the modified cells, once transplanted into the CNS of an aged rodent model, induced a significant increase in brain acetylcholine, brain-derived neurotrophic factor (BDNF), and glial-cell-derived neurotrophic factor (GDNF), improving animals’ cognitive function and physical activity [56].

#### 2.1.2. Skin Stem Cells in Wound Healing and Grafts

The adult mammal’s epidermis is a stratified squamous epithelium consisting of an inner layer of proliferative basal cells (keratinocytes) that are separated from the underlying dermis by a basement membrane. When the basal cells delaminate, they cease to proliferate and begin to differentiate, giving rise to the spinous, granular, and corneous layers. During the transition, following a series of morphological and biochemical changes, they culminate in the production of scales that are sealed to each other by lipid bilayers and form a body barrier. To constantly regenerate the epidermis, the scales are detached from the surface and replenished by internal differentiating cells that move outward and allow the frequent turnover replacing dying cells [57]. Adult skin epithelial progenitors are characterized by the expression of the keratin markers KRT5 and KRT14 [58,59]. These progenitors reside along the basement membrane separating the epithelium and dermal mesenchyme and are likely derived from both compartments. Progenitors attach and assemble the basement membrane via α6β4 and α3β1 integrins, which allows them to polarize [60,61]. In embryogenesis, after gastrulation, the epidermis exists as a single layer of unspecified epithelial progenitors, which begin to express progenitor-specific markers such as KRT5/KRT14. Subsequently, cells within the epidermal layer begin to produce higher WNT signals than other nearby cells, clustering into placodes [62]. At that point, if the underlying mesenchyme produces high levels of inhibitory bone morphogenetic protein (BMP), the high WNT-expressing cells form a hair follicle (HF), whereas if the underlying mesenchyme produces high BMP, the WNT-expressing cells form a sweat gland (SwG). All other cells that do not receive a morphogen gradient of the type described stratify to form the epidermis [63,64]. At the end of morphogenesis, the progenitors express the fundamental markers that identify them as keratinocyte progenitors. However, in the adult epidermidis, the progenitors have additional molecular features. Basal epidermal progenitors are primary respondents in the re-epithelialization during the wound healing process [65]. Dermal fibroblast progenitors differentiate into various cell types such as upper papillary fibroblasts (PF), lower reticular fibroblasts (RF), dermal condensate/dermal papilla (DP), and intradermal adipocytes/dermal white adipose tissue (DWAT), with specific functions such as HF formation and scar formation during wound healing. The heterogeneity and plasticity of dermal fibroblasts have recently been studied, which has implications for skin diseases and tissue engineering approaches [66]. Adult mammals do not have the ability to completely regenerate skin; however, the African spiny mouse, Acomys, is able to regenerate skin from wounds up to 60% of the body surface area, complete with follicles, glands, and DWAT, making it an emerging mammalian model [67].

A previous study reported that the fibroblast progenitor of the deeper skin connective tissue layer is marked by CD201 expression and controls wound healing by generating multiple specialized cell types, from proinflammatory fibroblasts to myofibroblasts, in a spatiotemporally regulated sequence. The identification of proinflammatory and myofibroblast progenitors and their differentiation processes provides new foundations for understanding and clinically treating wound healing [68]. The human embryo is able to heal skin wounds without scarring, a capacity that is lost around the third trimester of gestation likely due to the more mature response of the innate immune system and subsequent changes in adult skin expression [69,70]. Supporting the involvement of the immune system in scar formation is the study of oral cavity wounds that regenerate without scarring in adults and show a lower immune infiltrate and a lower transforming growth factor (TGF)-β1/TGF-β3 ratio compared to external skin wounds [71]. These results suggest that TGF-β isoforms regulate scarring and open the way to the use of anti-TGF-β drugs in fibrotic and wound healing contexts. TGF-β in the initial phases of wound healing is secreted by immune cells and platelets and later by fibroblasts [72]. Furthermore, it has been shown that the ratio of type 3 collagen to type 1 collagen is different in fetal wounds compared to postnatal wounds. Dermal fibroblasts in fetal wounds secrete more type 3 collagen than in postnatal wounds, in which there is a higher type 1 collagen expression. At the fetal level, this allows rapid, organized, and scar-free healing, unlike in postnatal wounds [73]. From these data, it can be deduced that changing gene expression in adult dermal fibroblasts to approximate that of the embryonic stage could provide new regenerative therapies for scar-free wound healing. Over the past 30 years, many studies have been conducted to develop epidermal stem cell (EPSCs)-based skin grafts to support acute and chronic wound healing in clinical dermatological and pharmacological applications [74]. The skin is an immunocompetent organ; therefore, the approach envisaged by clinical practice, to date, to permanently repair severe skin wounds is to use autologous skin grafts. The use of grafts for large skin lesions is limited by the amount of skin available for grafting, so research has focused on the use of EPSCs for skin regeneration [75] (Figure 1).

### 2.2. Mesoderm-Derived Tissues

Regarding the human mesoderm, a lot of information is still unknown because it forms in gestational weeks 2–4, when access to human embryos is inadmissible [76]. During gastrulation, a specific subset of pluripotent epiblast cells penetrate the primitive streak (PS) and spreads laterally between the ectoderm and visceral endoderm, leading to the formation of mesoderm. The formation of the PS marks the initial specification from which all mesodermal and endodermal tissue lineages arise. The canonical Wnt pathway is among several pathways that plays a crucial role in the gastrulation process: genetic studies in animal models have shown that embryos lacking Wnt3 and its co-receptors, or lacking the downstream effector β-catenin, are unable to establish the PS and therefore do not form the mesoderm [77]. Mesodermally derived organs such as kidneys, heart, and the axial skeleton develop from the maturation of mesodermal progenitors that, during the initial stages of gastrulation, differentiate into distinct mesodermal lineages (extraembryonic, lateral, intermediate, paraxial, and chordate mesoderm) [78]. The paraxial mesoderm is segmented into somites, constitutive elements of the trunk tissue, which are arranged along the dorsal–ventral axis to determine the ventral somite (sclerotome) that generates the bone and cartilage of the vertebral column and ribs, and the dorsal somite (dermomyotome), from which brown fat, skeletal muscle, and dorsal dermis are formed. The lateral mesoderm gives rise to the mesoderm of the limb buds and to the cardiac mesoderm, whose cells differentiate into cardiomyocytes and other heart constituents. Each line of differentiation is determined by the expression of specific genes [79]. The mesoderm forms the mesenchyme and the mesothelium. Mesenchyme is an embryonic precursor tissue from which several vertebrate organs and systems are formed: cartilage, bone, muscle, kidney, and the erythropoietic system. Specifically, mesenchyme originates from both the mesoderm and the neural crest through an epithelial–mesenchymal transition (EMT) of an ectodermal cell population. Therefore, ectodermal and mesodermal mesenchyme are formed in close proximity. This makes it difficult to understand the embryonic origin of some mesenchymal tissues, which is not very clear for all mesenchymal tissues [80]. The mesothelium is a coelomic epithelial structure derived from embryonic mesoderm. It plays an important role in the development of several organs, including the heart, lungs, and intestines, and is found covering a variety of organs and lining several different body cavities. In the adult, the mesothelium is a simple squamous epithelium that is found on the surface of all coelomic organs [81] (Figure 2).

#### 2.2.1. Human Adult Mesenchymal Stem Cells

Mesenchymal stem cells (MSCs) are multipotent cells able to differentiate into mature cells of several mesenchymal tissues, such as fat and bone. The discovery of MSCs occurred in the second half of the 1900s, when Friedenstein and his team identified the characteristic multipotency of these cells, but it was only in 1991 that Caplan et al. coined the term MSC. Over the years, numerous debates have arisen about the nomenclature of MSCs; some researchers prefer to define these cells as stromal cells. However, above all, their capacity for self-renewal justifies the definition of stem cells [82]. In 2006, Dominici et al. defined the minimum criteria that established the mesenchymality of a cell, which were listed in the Mesenchymal and Tissue Stem Cell Committee of the International Society for Cellular Therapy with the aim of promoting a more equitable characterization of MSCs. Specifically, MSCs must have the ability to grow and proliferate on plastic materials when cultured in vitro and differentiate into osteogenic, chondrogenic, and adipogenic lineages. MSCs are characterized by a fibroblastoid morphology and express specific mesenchymal markers such as CD105, CD90, and CD73, while they are negative for the expression of endothelial and hematopoietic surface markers CD11b, CD19, CD79α, CD31, CD34, and CD45 and human leukocyte antigen (HLA)-DR antigen [83]. MSCs are the most frequently described adult stem cell population and represent an ideal candidate for regenerative medicine. MSCs have paracrine functions that allow them to modulate inflammatory processes and immune responses [84]. The regenerative effects of MSCs are related to their multipotency, as well as their immunomodulatory/anti-inflammatory properties [85]. Studies in mouse embryos have shown that some MSCs originate from the neural crest [86]. Furthermore, Sox1+ neuroepithelial cells appear to be progenitors that give rise to MSCs through an intermediate neural crest stage [87]. Regarding the derivation of mesenchymal cells from mesoderm with the potential to differentiate into endothelial cells, blood cells, muscle cells, and cells of the mesenchymal lineage (bone and cartilage), the early mesodermal precursor with Flk1 positivity has been identified. However, the immediate mesodermal precursors that give rise to expandable multipotent MSC lineages are not fully characterized [88]. MSCs have been isolated from different adult tissues such as bone marrow and adipose tissue and represent a promising type of stem cells for cell therapy in new therapeutic approaches to regenerative medicine [89].

##### Bone Mesenchymal Stem Cells and Cell Therapies

Bone marrow was the first tissue in which MSCs were identified as progenitors of skeletal tissues [90]. Bone marrow mesenchymal stem/stromal cells (BMSCs) are multipotent cells, known as non-hematopoietic stem cells, that are in the medullary stroma of bone marrow and are characterized by their self-renewing, differentiating, and immunomodulatory properties [91]. BMSCs are plastic adherent cells and express CD105, CD73, and CD90 while lacking the expression of CD45, CD34, CD14, CD11b, CD79α, and CD19. STRO-1 is defined as a bone marrow mesenchymal stem cells marker, and it is highly expressed in young BMSCs while decreasing during expansion. Therefore, it is thought that the most suitable BMSCs for clinical application are STRO-1-positive ones [92]. The in vitro differentiation ability of BMSCs is determined by the following culture conditions: alkaline culture medium can stimulate the differentiation of BMSCs into chondrocytes [93], dexamethasone is required for osteogenic differentiation [94], bone morphogenetic proteins (BMPs) can induce BMSCs into osteocytes through the expression of Runt-related transcript 2 (Runx2), and azacitidine can enhance the osteogenic and adipogenic differentiation ability of BMSCs in 3D cultures [95]. BMSCs are able to transdifferentiate into other cell types and dedifferentiate back into a undifferentiated state in response to certain extracellular signals [96]. BMSCs are the most commonly used in stem cell therapies because they play an important role in tissue healing through migration, immune regulation, and differentiation at sites of damaged tissue [97,98]. In 2009, adult non-hematopoietic stem cells were first isolated from the bone marrow and transplanted into patients [99]. BMSCs are used for tissue engineering and regenerative medicine, but they are represented in very low levels in bone marrow tissue, so before their use, BMSCs have to amplify 500-fold higher in 50 passages during cell culture [100]. Cell therapies with BMSCs have shown promising results in the treatment of neurological diseases. In a 2017 study, autologous BMSCs were injected intrathecally into 26 patients with amyotrophic lateral sclerosis (ALS), leading to a slowing of the disease development without significant adverse events [101]. BMSCs have shown the ability to differentiate into the endodermal lineage, such as hepatocyte-like cells. Some clinical studies have demonstrated the efficacy and feasibility of BMSC therapy in patients with liver diseases like liver cirrhosis, where transplantation led to improvement of the disease [102]. In 2021, allogeneic BMSC transplantation in patients with acute kidney injury was shown to improve tissue repair [103]. Intramyocardial infusion of autologous BMSCs in conjunction with transmyocardial revascularization or coronary artery bypass grafting showed improved regional contractility in cell-treated regions at 1-year follow-up and decreased angina at 12 months post-treatment [104]. Falanga et al. demonstrated that cell therapy with autologous BMSCs represents a safe and efficient method for wound healing, leading to complete wound closure and tissue regeneration within 4–5 months without adverse events [105]. BMSCs in the treatment of various bone abnormalities and defects represent a promising, safe, and effective strategy for bone regeneration, since in several studies, it has been demonstrated that treatment with these cells significantly improves the clinical manifestations of patients with different bone diseases [106]. Several clinical studies have shown that BMSCs can promote alveolar and maxillary bone regeneration and improve craniomaxillofacial bone defects [107].

One of the main limitations is that the functional capacities of BMSCs decrease with aging, so their use is strongly age-dependent. The age factor of the donor in cell-based therapies becomes limiting for older patients [108].

##### Adipogenic Mesenchymal Stem Cells: Differentiation and Clinical Applications

Adipose tissue is derived from mesoderm during embryonic development. Adipose tissue can be essentially divided into two types: white and brown. White adipose tissue is mainly involved in the storage of excess energy in the form of lipids and represents the source of adipogenic stem cells (ASCs) in most studies [109]. The existence of ASCs within brown adipose tissue has also been demonstrated, but compared to ASCs derived from white adipose tissue, brown adipose-derived stem cells (ADSCs) have a greater capacity to differentiate into active brown adipocytes while showing similar multilineage differentiation potential of white ADSCs [110].

In 2001, Zuk et al. published on the plasticity of pre-adipocytes and their stemness characteristics, such as self-renewal, asymmetric division, and multipotency, which led to their identification as “adipose-derived stem cells” [111].

ASCs have the morphological and immunophenotypic characteristics of MSCs [112]. The advantage of using ASCs for regenerative medicine is the minimally invasive collection procedure in high quantities without ethical problems. ASCs are characterized by specific cell surface markers. They are positive for CD13, CD29, CD34, CD44, CD73, CD90, CD105, CD166, MHC class I, and HLA-ABC expression and negative for CD38, CD45, CD106, HLA-DR, DP, DQ (MHC class II), CD80, CD86, CD40, and CD40L (CD154) [113]. CD34 reaches higher levels in early passage ASCs, while its expression decreases during the culture period [114]. ASCs not expressing MHC-II are able to inhibit the proliferation of activated peripheral blood mononuclear cells, modulating the immune system response in inflammatory disorders or in allogeneic transplantation [115].

ASCs may be isolated from human adipose tissue and are characterized by a high potential to differentiate into mature adipocytes and into other cells of mesenchymal lineages, including osteoblasts, chondrocytes, and cardiac muscle. For these reasons, these cells are a valuable source for clinical applications in the field of regeneration and clinical reconstruction, especially of soft tissues [116]. In plastic surgery, autologous fat grafting is becoming increasingly popular for soft tissue augmentation, rejuvenation, and reconstruction [117].

In 2006, a technique called cell-assisted lipotransfer (CAL) was developed, which involves combining aspirated fat with ASCs to create stem-cell-rich fat grafts [118]. These grafts were transplanted into mice with severe immunodeficiency, showing a significantly higher survival rate and better angiogenesis than those transplanted with conventional fat grafting. CAL with ASCs has been used for facial rejuvenation and breast augmentation [119,120].

ASCs undergo osteogenic differentiation by regulating osteogenesis-related signaling of Wnt, TGF-β, PI3K/AKT, MAPK, Hippo, and JAK-STAT pathways, positioning them as optimal progenitor sources for bone tissue engineering [121]. ASCs can also differentiate into cells of non-mesenchymal lineages, such as endothelial, myogenic, and neuronal lineages, following treatment with a combination of different inductive factors [122].

However, the therapeutic potential of ASCs is not limited to cell replacement, but they also have paracrine activity by secreting a broad spectrum of bioactive molecules (cytokines, antioxidant factors, chemokines, and growth factors) that can mediate therapeutic effects, including facilitation of angiogenesis, suppression of apoptosis, and participation in immunoregulation [123]. In detail, ASCs secrete vascular endothelial growth factor (VEGF), TGF-β, platelet-derived growth factor (PDGF), angiogenin (ANG), and other angiogenic cytokines that stimulate the angiogenic process and facilitate tissue regeneration [124]. In contrast, their anti-apoptotic effect is mediated by the secretion mainly of insulin-like growth factor-1 (IGF-1) [125]. Immunoregulation is determined by the ability of ASCs to suppress the differentiation of dendritic cells; the synthesis of immunoglobulins, CD8+, and CD4+ T lymphocytes; and the proliferation of natural killer cells, promoting instead the polarization of M2 macrophages and the proliferation of regulatory T cells [126]. The skin exhibits self-healing and renewal functions, but the wound healing process can be compromised in particular circumstances, such as deep burns or in pathological contexts like diabetes mellitus [127]. For refractory wounds, ASCs have been identified as potential therapeutics through their paracrine effects and differentiation ability into skin cells. ASCs can migrate to wound sites and differentiate into endothelial cells, dermal fibroblasts, and keratinocytes or promote wound healing via paracrine pathways [128,129,130]. ASCs have shown the ability to differentiate into the myogenic lineage through a myogenic medium, demonstrated by the expression of the muscle-specific markers MyoD and myosin heavy chain. ASCs have higher myogenic potential when pretreated with interleukin (IL)-4 and stromal cell-derived factor (SDF)-1 [131,132]. It was observed that although both ASCs and BMSCs improved the healing of skeletal muscle tears, the use of ASCs showed less fibrosis, identifying ASCs with a more promising therapeutic potential than BMSCs in the treatment of skeletal muscle injuries [133]. Encouraging results have also been reported for the induction of ASCs to tenogenic differentiation, although there is no standardized protocol to date. Rao et al. prepared a soluble extracellular matrix from bovine tendon (tECM) and used it to treat ASCs. After such treatment, cells showed increased proliferation and improved tenogenic differentiation [134]. In a clinical study, 44 patients with degenerative rotator cuff tears, after undergoing arthroscopic rotator cuff repair, were injected with lipoaspirate tissue containing autologous ASCs. The lipoaspirate tissue effectively promoted the functional repair of the rotator cuff [135].

ASCs therapy has also been extensively studied as a promising strategy for the treatment of myocardial infarction. Spontaneous ASCs cardiomyogenic differentiation was observed for the first time in the semisolid medium of methylcellulose, where the cardiomyocyte-like cells expressed specific cardiac markers and showed pacemaker activity [136]. Kastrup et al. have injected allogeneic ASCs into the infarct border zone in 10 patients with ischemic heart failure, finding that left ventricular ejection fraction and exercise capacity improved [137]. Similarly, in another clinical study, intramyocardial injections of autologous ASCs improved cardiac symptoms [138].

ASCs are promising tools for nerve regeneration. The neuronal microenvironment and the extracellular matrix play a crucial role in promoting myelin formation by stem cells. ASCs differentiated towards Schwann cells (dASCs) mimic various functional capabilities of Schwann cells. In dASCs, laminin appears to promote the release of nerve growth factor (NGF), which, being a regulator of axonal elongation, could be a suitable candidate for nerve sheath guide conduits for targeted approaches in nerve injury treatment with ASCs [139]. Huang et al. demonstrated that treatment of ASCs with fibroblast growth factor 9 (FGF9) can induce their differentiation into Schwann cells via the FGF9-FGFR2-Akt pathway. When these differentiated cells were transplanted into a rat sciatic nerve injury model, they led to the formation of the myelin sheath, promoting nerve regeneration [140]. The neuroregenerative potential of ASCs is actually explained by the “paracrine hypothesis”: ASCs secrete neurotrophic factors that in turn stimulate the neurotrophic factors’ secretion by Schwann cells, leading to improved myelination, regeneration, and a reduction in nerve fibrosis [141].

Despite the recent encouraging results of preclinical and clinical studies on ADSCs, ASC transplantation could increase the risk of tumor growth and metastasis since the properties that make ASCs a good source for tissue regeneration (angiogenesis, cell homing, and immune regulation) are the same that could induce tumor progression. ASCs can be recruited into tumors and integrated into the tumor stroma; then, some cells remain ASCs, while other cells are converted into cancer-associated fibroblasts [142,143].

##### Oral Mesenchymal Stem Cells as an Alternative Source for Regenerative Medicine

As mentioned above, the most well-known sources of MSC are bone marrow and adipose tissue. The limitation especially of BMSC in therapeutic use is the relatively low percentage of cells and the medical intervention required for their collection [144]. For this reason, alternative and more easily available MSCs sources have been identified, such as oral cavity tissues, which include dental pulp, apical papilla, dental follicle, gingiva, and periodontal ligament [145]. The MSCs isolated from oral cavity tissues showed the ability to adhere to plastic culture dishes and to expand through consecutive in vitro passages without any modifications in the stemness profile [146]. Compared to MSCs isolated from other tissues, oral MSCs not only have the same mesenchymal characteristics but are also easier to obtain clinically (discarded tissues) and have a better proliferation rate, cellular function, homogeneity, and lower tumorigenicity [147]. Oral MSCs, like other MSCs, are multipotent non-hematopoietic adult stem cells, which showed positive expression for surface markers CD73, CD90, and CD105 and negative expression for hematopoietic surface markers, such as CD14, CD34, CD45, HLA-D-related surface molecules (HLA-DR or MHC-II) [148]. Morphologically, oral MSCs have fibroblastoid appearance with long cytoplasmic processes and many filopodia [149,150].

Dental pulp is an important component of the dental body that serves to maintain tooth life and function. Human MSCs extracted from the pulp (hDPSC) have the capacity for odontogenic differentiation, posing as an alternative clinical therapeutic option compared to endodontic treatment for dental pulp repair and regeneration [151]. hDPSCs show gene expression profiles and differentiation capacity similar to BMSCs but are easier to find because they are routinely extracted with dental procedures [152]. hDPSCs originate from the neural crest and have high proliferative activity and the ability to differentiate into multiple cell types. They also show osteogenic and dentinogenic potential, making them ideal candidates for autologous regenerative therapies of various dental structures, including the dentin–pulp complex and periodontal tissues [153]. Through modulation with specific factors, hDPSCs can differentiate into different cell types such as odontoblasts, osteoblasts, chondrocytes, cardiomyocytes, neuronal cells, and adipocytes [154]. hDPSCs have been shown to repair periodontal tissue, ischemic tissue in diabetic limbs, bone damaged by osteonecrosis, skin burns, liver, neuronal tissue, skeletal muscle tissue, and blood vessels. For these reasons, DPSCs are considered one of the best future sources of MSCs for regenerative medicine [147]. In a myocardial infarction animal model, hDPSC transplantation improves cardiac function by inducing angiogenesis and reducing the infarcted area [155]. Furthermore, hDPSCs could be an ideal choice for the regeneration of severed nerves during surgery or following trauma since DPSCs have been shown to release factors that mediate axonal guidance in the trigeminal nerve [156]. Human periodontal ligament stem cells (hPDLSCs) reside in the perivascular space of the periodontium and possess MSCs characteristics. They represent a promising tool for regeneration with great progress in hPDLSC transplantation [157]. hPDLSCs show the ability to differentiate into mesengenic lineages. They are characterized by immunomodulatory properties which allow them to protect against infectious diseases with an active role in the immune response, thanks to the interaction with immune cells. hPDLSCs avoid the improper activation of T lymphocytes and modulate the immune response during healing processes [158]. hPDLSCs are a promising regenerative autologous source. They are readily available, stable, and have excellent biological properties such as immune modulation and neuroprotection. hPDLSCs can be used in neural tissue repair since they originate from the neural crest and therefore have high potential to differentiate into various neural cells, as well as release neurotrophic factors that can aid in myelination and neuron growth [159]. hPDLSCs are considered promising for regenerative therapy in the periodontium; however, their rarity slows the progression of basic and clinical research [160].

Gingiva is one of the four pillars of connective tissue that surrounds the tooth and attaches to the alveolar bone, creating the gingival attachment and supporting the tooth. Human gingiva has greater regenerative capabilities than skin, with gingival stem cells rapidly regenerating the tissue, healing wounds with little or no scarring [161,162]. hGMSCs can derive partly from the neural crest and partly from the mesoderm [163].

The same features of other oral MSCs are shown by MSCs derived from human gingiva (hGMSCs), which are capable of differentiating into mesengenic lineages and with immunomodulatory properties by directly interacting with natural killer cells, dendritic cells, B lymphocytes, and T lymphocytes [164]. Several experiments have demonstrated that, in the presence of appropriate induction factors, hGMSCs can differentiate into both mesodermally derived cells (osteoblasts, chondrocytes, and adipocytes) and ectoderm-derived cells (neurons, odontogenic cells, and keratinocytes) [165]. hGMSCs have been studied as a medium for repairing craniomaxillofacial bone defects. In vivo studies of critical-sized mandibular and calvarial bone defects in rats have shown that GMSCs not only promote new bone formation but, when transplanted systemically, can target the mandibular defect site and promote bone regeneration [166,167]. El-Latif et al. demonstrated that GMSCs injected directly into the cut area of submandibular salivary glands significantly improved the regeneration of ductal, acinar, and myoepithelial cells compared to the use of autologous fibrin glue alone to close the cut [168]. Several studies have investigated the potential of GMSCs for nerve regeneration. Under certain culture conditions, hGMSCs can be induced into neural crest-like stem cells (NCSCs) characterized by increased expression of NCSC-related genes, and they showed a high capacity to differentiate into Schwann-like cells. NCSCs derived from hGMSCs, when transplanted into a facial nerve defect rat model, facilitated facial nerve regeneration and functional recovery [169]. In induced animal models of Parkinson’s disease, hGMSC injection improved the pathology through an anti-apoptotic mechanism mediated by the BCL-2/BAX pathway [170]. hGMSCs also have regenerative effects on the skin: through the EGFR/STAT3 signaling pathway, hGMSCs promoted proliferation, migration, and DNA damage repair capacity of skin cells following injury [171]. hGMSCs show myogenic differentiation potential. Subcutaneous transplantation of hGMSCs encapsulated by a specific scaffold with multiple growth factors led to a cell morphology change toward that of typical muscle cells and also expressed gene markers related to muscle regeneration. In mouse models, transplantation of GMSCs showed a greater capacity for myogenic regeneration compared to BMSCs [172].

##### Menstrual-Blood-Derived Stem Cells: Resource Free from Ethical Concerns in Clinical Perspectives

Menstrual-blood-derived stem cells (MenSCs) are a novel source of MSCs that were discovered in 2007 by Meng et al. and Cui et al. [173,174]. MenSCs are extracted from menstrual blood, and their original sources are deciduous endometrial stem cells [175]. A comparison study of six sources of MSCs suggests that MenSCs possessed higher proliferation rates, painless procedures, and almost no ethical issues [176]. MenSCs are similar to BMSCs in phenotypes, classical three-line differentiation, and surface marker expression [175].

MenSCs express CD9, CD29, CD44, CD73, CD90, CD105, octamer binding transcription factor 4 (OCT-4), CD166, MHC I, and C-X-C chemokine receptor 4 (CXCR4). Therefore, CD29, CD73, CD90, and CD105 are in common with the expression of MSC markers, also sharing with MSCs the negativity for hematopoietic stem cell markers such as CD34, CD45, CD133, and for HLA-DR [177]. Currently, the potential benefits of using MenSCs for clinical practice are increasingly being understood, and although few clinical studies have been performed yet, their therapeutic potential has already been recognized in several types of diseases in preclinical research.

It was found that MenSCs are able to induce angiogenesis in an in vivo matrigel plug assay and can support endothelial cell growth by stimulating expansion more than BMSCs [178]. Hida et al. found that potent cardiac precursor cells can be obtained from MenSCs, since, with specific induction factors, MenSCs started to beat spontaneously, exhibiting cardiomyocyte-specific action potentials. This provides a new non-invasive approach to cell therapy for cardiac diseases [179]. To gain new tools for the treatment of reproductive system disorders, the potential of MenSCs was analyzed, and it was demonstrated that they are able to form oocyte-like cells with the expression of related germline markers [180]. With specific culture media and the right concentrations of hepatocyte growth factor and oncostatin M, MenSCs can differentiate into functional hepatocyte-like cells. This opens new avenues for liver regeneration [181]. MenSC transplantation has been tested on animals with different pathological conditions, demonstrating that MenSC transplantation leads to an improvement in liver fibrosis [182], in diabetes with the β-cell regeneration [183], in ischemic stroke [184], in skeletal muscle repair caused by Duchenne dystrophy [174], in Alzheimer’s disease [185], in skin regeneration, and in wound repair [186]. All these pre-clinical studies lay the basis for future clinical strategies exploiting MenSCs’ potential. Despite the progress, the limitations related to the potential use of MenSCs concern the lack of information regarding the survival time of MenSCs in foreign bodies and, above all, the lack of data that guarantee their long-term safety [187,188].

#### 2.2.2. Hematopoietic Stem Cells and Transplantation

Hematopoietic stem cells (HSCs) are a rare cell population with the capacity for self-renewal and differentiation to give rise to all blood cell lineages. Definitive HSCs emerge from the hemogenic endothelium within a region of embryonic mesoderm called the aorta–gonad–mesonephros and then travel via the circulation to the fetal liver before colonizing the adult bone marrow [189]. In detail, HSCs in adult bone marrow arise from the replication of the HSC pool that emerged early in ontogeny, when the bone marrow had not yet formed. In vertebrates, the first hematopoietic activity is represented by the appearance of blood islands in the extraembryonic yolk sac mesoderm, which supports primitive erythropoiesis. Erythropoiesis is subsequently transmitted from the liver and spleen and finally from the thymus and bone marrow, where postnatal hematopoiesis is established [190]. HSCs differentiate into mature lineage cells through a continuous process in which restricted unilineage cells emerge directly from a continuum of undifferentiated hematopoietic stem and progenitor cells (HSPCs), without major transitions through the multi- and bi-potent phases [191]. Human HSCs can be isolated from various sources, including bone marrow, the umbilical cord blood, or peripheral blood, with specific strategies and kits like microbeads or flow cytometry. HSPCs isolated from bone marrow have been successfully used for 50 years in hematological transplantation. Currently, these cells are most frequently isolated from mobilized peripheral blood or umbilical cord blood [192]. Recently, the existence of human kidney-derived HSCs capable of self-renewal and multilineage hematopoiesis has been discovered: in a pediatric kidney transplant, kidney-derived HSCs took up long-term residence in the recipient’s bone marrow and gradually replaced their host counterparts, leading to blood group conversion and full donor chimerism of both lymphoid and myeloid lineages [193]. The most common strategy for isolating HSCs is the lineage-negative marker (Lin−)CD34+CD38−CD45RA−CD90+CD49f+, but recent studies have identified the evaluation of endothelial protein C receptor (EPCR) instead of CD90 and have added G-protein-coupled receptor family C group 5 member C (GPRC5C) to distinguish dormant from active HSCs, leading to a new identification strategy: Lin−CD34+CD38−CD45RA−CD49f+EPCR+. These markers allow the identification of stem cell frequency of approximately one in five cells [194]. HSC transplantation represents the most widely used regenerative therapy. The entire hematopoietic system can be regenerated through HSC transplantation, which allows the treatment of various hematological diseases. Each year, more than 30,000 patients with hematologic malignancies, following treatment with high-dose chemotherapy, are transplanted with bone marrow HSCs. Finding HLA-matched bone marrow is difficult, so with the increased availability of stored umbilical cord samples, umbilical cord is moving to be the most important source of HSCs for transplantation, although it is largely limited to pediatric cases because a single cord usually does not contain enough HSCs for an adult [195]. HSCs influence the bone repair process, mainly through the promotion of osteogenesis and angiogenesis at the injury site; therefore, HSCs co-cultured with other bone cell types represent a powerful tool for tissue engineering strategies that promote bone regeneration and healing [196]. Over the past 10 years, several clinical studies have demonstrated that HSC transplantation can have a therapeutic effect, even in diseases of organs other than hematopoietic origin, including the liver, heart, and brain. In the field of liver regeneration, HSCs have been shown to give rise to hepatocyte-like cells after in vivo transplantation, completely regenerating the liver of FAH mice and correcting the disease [197,198]. These studies have also been subsequently confirmed in other types of rodents, demonstrating the hepatocytic potential of HSCs from various sources in various mouse models with a wide range of genetic lesions and lesions induced by chemical or physical means [199]. Eglitis and Mezey [200] demonstrated that murine HSCs were able to differentiate into both microglia and astroglia, but subsequent in vitro studies have provided contradictory evidence that murine bone-marrow-derived HSCs do not possess the ability to form functional neuronal cells [201]. In one study [202], it was observed that HSC transplantation resulted in a semi-selective accumulation of HSCs at the lesion site, with a reduction in the volume of the ischemic lesion. However, current studies on HSC differentiation into neuronal cells are too few to conclude on an HSC-mediated improvement in ischemic injury/stroke clinical settings. Cardiac regeneration is probably the area in which HSCs have been most extensively studied as a regenerative approach for myocardial infarction. Numerous preclinical studies have been conducted to evaluate the cardiomyogenic potential of HSCs, and with the exception of one study, it has been demonstrated that infusion of HSCs or administration of granulocyte-colony-stimulating factor (G-CSF) to mobilize the patient’s own endogenous HSCs can result in a reduction in the infarct size with functional improvement [203].

### 2.3. Endoderm-Derived Tissues

Endoderm is the innermost germ layer of embryos, surrounded by mesoderm and ectoderm. The definitive endoderm is composed of a multipotent stem cell population formed during gastrulation. The endoderm gives rise to the main cell types that constitute many internal organs (thyroid, thymus, lung, stomach, liver, pancreas, intestine, bladder, prostate, and urethra) [204]. After gastrulation, a series of morphogenetic movements cause the endoderm to become a primitive intestinal tube surrounded by mesoderm. The intestinal tube has a dorso–ventral (D-V) and an antero–posterior (A-P) region that allows one to distinguish the foregut, midgut, and hindgut domains, whose formation is dictated by molecular processes with different gene expression. Molecular profiles are increasingly refined in precise domains in which specific organs will form. More specifically, the foregut gives rise to the esophagus, trachea, stomach, lungs, thyroid, liver, biliary system, and pancreas; the midgut forms the small intestine; and the hindgut forms the large intestine. Organ buds develop as outgrowths of the endodermal epithelium that, mixing with the surrounding mesenchyme, differentiate after proliferation during fetal development into functional organs. Endodermal organogenesis processes are controlled by many growth factor pathways, including FGF, BMP, WNT, retinoic acid (RA), Hedgehog, and Notch [205]. The study of endoderm-derived tissue development is fundamental for tissue regeneration in diseased adult organs through cell transplantation therapies or organ endogenous stem cell stimulation (Figure 3).

#### 2.3.1. Intestinal Stem Cells and Intestinal Regeneration

Given the constant exposure to environmental insults, the intestinal epithelium has a remarkable regenerative capacity, completely self-renewing every 3–5 days. This high regenerative capacity is maintained by intestinal stem cells (ISCs) [206]. Adult ISCs are located at the bottom of the crypts of Lieberkühn, where they are responsible for constantly supplying the intestinal epithelium with new cells. ISCs express specific markers such as Leucine-Rich Repeat Containing G Protein-Coupled Receptor 5 (LGR5) [207]. The key mitogen responsible for intestinal epithelial renewal is Wnt, which targets the LGR5 pathway gene and drives intestinal renewal. LGR5-positive cells can be traced back to all differentiated intestinal epithelial cell types from the duodenum to the colon [208]. LGR5+ cells produce absorptive and secretory progenitor cells that are pushed out of the crypt base and undergo further rounds of cell division as they migrate along the crypt wall. This leads to progressive differentiation of the progenitor cells into the mature epithelium postmitotic cell types. The differentiated cells continue their migration to the tip of the villus, where they are released into the lumen [209].

The secretory Paneth cell of the small intestine is the only differentiated cell type that does not undergo this ascending process and instead remains in the crypt [210].

Intestinal transplantation is still accompanied by many limitations, including immune rejection and donor availability. Stem cell therapy and tissue engineering appear to be promising tools using ISC for the regeneration of damaged intestine. In addition to Lgr5, surface protein markers typical of ISCs and that could allow their isolation by FACS are Doublecortin-like kinase 1 (DCAMKL1), polycomb ring finger Bmi-1, and CD24 [211].

ISC transplants have been tested in vivo by several researchers [212]. Already in 1994, Tait et al. demonstrated that, in a rat model in which a surgical mucosectomy was performed on the intestinal segment of the colon, ISC transplantation led to tissue regeneration after 2 weeks [213]. In a very recent study, ISCs from healthy adult rats were transplanted into stroke rat models. ISCs from young adults (of both sexes) transplanted into middle-aged animals of the same donor’s sex led to the repair of intestinal tissue and reduced intestinal permeability. The transplantation also affected the animals’ behavior, preventing depressive behaviors and age-related cognitive decline, while ISCs from middle-aged adult donors did not improve the pathological conditions, demonstrating that donor age is a determining factor for transplantation [214].

#### 2.3.2. Liver-Derived Stem Cells and Their Use as Treatment in Liver Diseases

Liver stem cells (LSCs) are adult stem cells present in the liver, mainly hepatic oval cells (HOCs) and small hepatocyte-like progenitor cells (SHPCs), which can differentiate into hepatocytes or bile duct epithelial cells, regenerating hepatic tissue [215]. Recently, LSCs have been identified as a proliferating and self-renewing cell population localized adjacent to the central vein in the liver lobule, where they can differentiate into hepatocytes, replacing all hepatocytes along the liver during homeostatic liver renewal [216]. The liver arises from the anterior portion of the definitive endoderm [217]. Mouse organogenesis experiments have revealed the presence of three distinct domains of hepatic progenitor cells located in the medial and bilateral regions of the foregut [218]. Among the transcription factors that control the early stages of liver development is albumin, which is one of the best characterized markers of nascent liver cells. Forkhead Box A2 (FoxA) and GATA Binding Protein 4 (GATA4), expressed in the anterior endoderm, initiate albumin expression by binding to its enhancer [219]. Although few specific markers are available, and some of them are in common with hematopoietic progenitors, recently, LSCs have been well characterized and isolated with FACS on the basis of the following non-specific cell surface markers: c-Kit, CD45, TER119, c-Met, and Dlk [220]. Many LSC surface markers are in common with hematopoietic progenitors because hematopoiesis and liver development share common stages during fetal development: hematopoietic stem cells exit the yolk sac and move into the developing liver, which remains hematopoietic throughout the entire fetal period until the first week after birth in neonates [221].

It is estimated that the liver is replaced by normal tissue renewal approximately once a year thanks to LSCs [222].

Due to their ability to directionally differentiate into hepatocytes or cholangiocytes, LSCs can be considered as an ideal cell therapy for the treatment of liver diseases, even without ethical issues. Several studies demonstrate the in vitro differentiation ability of LSCs into liver cells and their potential as a regenerative strategy [223].

In murine models of acute liver injury, concanavalin A-induced, intraperitoneal transplantation of human-derived LSCs demonstrated their protective effect on the injured tissue by modulating the regulatory T cells and T helper 17 cells [224].

Furthermore, in mice models with carbon tetrachloride (CCl4)-induced fibrosis, weekly transplantation of human liver stem cells (LSCs) was shown to prevent the progression to advanced stages of fibrosis, including bridging fibrosis and cirrhosis. All analyzed liver function markers (especially alanine aminotransferase and aspartate aminotransferase) were decreased in transplanted mice, suggesting an improvement in liver function and repair of damage [225]. In one study, stem cells derived from human fetal liver were used for transplantation in patients with end-stage liver cirrhosis. In the 25 patients with liver cirrhosis considered, the LSC transplantation led to a clear clinical improvement in terms of all clinical and biochemical parameters, offering new approaches to end-stage liver diseases [226].

## 3. Embryonic Stem Cells

Embryonic stem cells (ESCs) are pluripotent cells that give rise to all types of somatic cells in the embryo. They arise from the blastocyst, which is formed following multiple mitotic cell divisions of the zygote (totipotent) during early embryogenesis. The blastocyst contains an inner layer of cells called the embryoblast, also known as the inner cell mass (ICM), and an outer layer called the trophoblast. The trophoblast gives rise to extraembryonic tissue, including the placenta, chorion, and umbilical cord, while the ICM develops the three germ layers from which the entire embryo with future adult tissues is formed [227]. ESCs, given their pluripotency, would be the best solution for cell therapy in regenerative treatments, but the ESCs’ use for research purposes still generates ethical and legal debates [228].

For this reason, in addition to the study of stem cells described above, the research has identified a way to obtain pluripotent cells without using embryonic ones: induced pluripotent stem cells (iPSCs).

## 4. Induced Pluripotent Stem Cells: The Future of Personalized Regenerative Medicine

iPSCs are obtained from somatic cells that are reprogrammed into pluripotent stem cells by introducing a combination of different transcription factors, such as Oct3/4, Sox2, Klf4, and c-Myc, which induce the pluripotency stage [229]. Since iPSCs from a patient can potentially differentiate into any adult cell, they could be a useful source for cell transplants as well as models for studying physiopathological mechanisms in the identification of new therapies.

iPSCs were first generated from murine fibroblasts in 2006, with the demonstration that the resulting cells had ESC properties [230].

iPSCs are in an embryonic stem-cell-like state through the forced expression of genes and factors necessary for maintaining the embryonic properties. iPSCs are embryonic-like in many ways: the expression of ESC markers, chromatin methylation patterns, embryoid body formation, teratoma formation, viable chimera formation, pluripotency, and the ability to differentiate in vitro into any adult tissue. iPSCs represent the most promising tool for personalized regenerative medicine since tissues derived from iPSCs will be nearly identical to the cell donor, which is essential for disease models and drug screening [231]. There are several methods to obtain iPSCs. Starting from the oldest, the reprogramming methods can be summarized as nuclear transplantation, cell fusion, reprogramming by cell extracts, and direct reprogramming by gene manipulation. Direct reprogramming can be further divided into two methodologies: vector integration methods and non-integrative genomic methods. Direct reprogramming is the most widely used method to generate iPSCs today, and specifically, non-integrative methods ensure greater stability of the cell line generated [232]. iPSCs can be obtained from various types of donor cells. Fibroblasts are the most commonly used somatic cell type for the generation of iPSCs since they can be easily obtained via punch biopsy and are easily cultured. The limitation is that punch biopsy is an invasive approach and fibroblasts have also shown lower reprogramming efficiency compared to other cell types [233].

iPSCs have also been obtained starting from epidermal keratinocytes and renal tubular epithelial cells. The limitation of these sources is the poor acquisition of cellular immortality, which limits their expansion in culture and their use [234,235,236]. Given the easy availability of blood cells with routine blood collections, these cells have been used for reprogramming into iPSCs. So far, several types of cells isolated from blood have been used as donor cells to obtain iPSCs [237,238]. CD34+ stem cells have been the most efficient source for inducing pluripotency, but their isolation and mobilization by subcutaneous G-CSF injection in patients represent important limitations [237]. T lymphocytes have also been shown to reprogram efficiently, but the resulting iPSCs have pre-existing V(D)J rearrangements at T cell receptor loci that can lead to the T cell lymphoma development [238,239,240]. In a very recent study, a new source of iPSCs has been identified in gingival mesenchymal stem cells, which have been reprogrammed by a non-integrating method, generating an autologous stable iPS line capable of giving rise to embryoid body formation and to the differentiation of the three germ layers. In the same study, it was also demonstrated that reprogramming not only induced a change in gene and protein expression toward the pluripotent phenotype but also induced a rearrangement of membrane CD expression compared to the parental cell line and an epigenetic change, since the extracellular vesicles produced by iPSCs contained an miRNA content with different expression compared to the miRNA content of the vesicles produced by the parental line, with eight miRNAs appearing de novo in the iPSC vesicles that were not present in the parental ones [241]. In the iPSC-based 3D modeling of in vitro diseases, it is important to consider the phenotyping of these cells for better identifying molecular events that lead to the development of 3D organoids and the study of the disease. Manganelli M. et al. demonstrated that reprogramming of CD34+ progenitor cells and skin fibroblasts leads to the expression of the progesterone receptor (PR), providing crucial information for using iPSCs as an in vitro modeling tool [242].

Several reseachers have begun to deposit novel patient-specific human iPS lines for therapeutic and regenerative purposes. Park et al. generated iPSCs derived from patients with a variety of conditions, including Huntington’s disease, muscular dystrophy, diabetes mellitus, Down syndrome, and others [243]. In particular, iPSCs derived from patients with neurodegenerative diseases, differentiated in vitro into the neuronal cell type involved, provide for the first time in vitro models to study the onset, cellular characteristics, and contribution of environmental factors to the diseased phenotype [244].

Since iPSCs share a number of characteristics with tumor cells, the generation of iPSCs from human tumor cells represents an opportunity to develop in vitro models of carcinogenesis, which is especially important for tumors such as glioblastoma and gastrointestinal cancer, whose lack of a relevant model has so far limited translational research [245].

One of the limitations of human skin regeneration is obtaining an unlimited number of skin dermal cells. In one study, iPSCs were differentiated into skin-derived precursor cells (SKPs) to reconstitute the skin dermis. Human iPS-derived SKPs showed similar gene expression characteristics to SKPs isolated from human skin dermis and were able to differentiate into neural and mesodermal progenies, including adipocytes, skeletogenic cell types, and Schwann cells, providing novel therapeutic approaches [246].

Skin fibroblasts from a child with spinal muscular atrophy (SMA) were reprogrammed to iPSCs. These cells, expanded in culture, retained the disease genotype and, when differentiated into motor neurons, showed selective deficits not present in motor neurons derived from the child’s unaffected mother. This demonstrates that human iPSCs can be used to develop models of genetically inherited diseases, allowing the study of their pathophysiological mechanisms and the identification of new therapies [247].

In culture, iPSCs can differentiate into several neuronal and glial cell types. In mouse models, transplantation of iPSCs into fetal mouse brains has shown that these cells differentiate into glia and neurons, including glutamatergic, GABAergic, and catecholaminergic subtypes. Furthermore, iPSCs were induced to differentiate into a dopaminergic phenotype that, when transplanted, was able to improve behavior in a rat model of Parkinson’s disease (PD), highlighting their therapeutic potential [248].

For bone tissue regeneration purposes, different protocols have been set up to differentiate iPSCs into osteoblasts and osteocyte-like cells, leading to excellent results [249].

In 2009, Nelson et al. first investigated the efficacy of iPSs in the treatment of acute myocardial infarction, demonstrating that iPSCs led to the in situ regeneration of cardiac, smooth muscle, and endothelial tissue [250].

iPSs also represent a starting point for obtaining autologous hematopoietic cells. In one study, iPSs were shown to generate hematopoietic stem cells through the ectopic expression of specific transcription factors, providing a new personalized source for transplantation [251].

Recently, it has been demonstrated that iPSCs obtained by reprogramming human endometrial stromal cells are able to differentiate into hematopoietic and erythroid lineages through a two-phase culture system, identifying new tools for tissue regeneration [252] (Figure 4).

## 5. Conclusions

Despite the important progress brought about in regenerative medicine by the use of different adult stem cells, there are still important limitations related to their use. Unipotent adult stem cells represent a limited source that has given way to MSC research, but despite the various advantages, including the capacity for trans-differentiation, the administration of fully differentiated MSCs is associated with a higher risk of immune rejection. MSCs are not able to differentiate into all types of tissues. Therefore, the identification of new tools such as iPSCs has become necessary and allows us to overcome the previously described limitations related to stem cells, freeing research from the ethical problems of using ESCs. By generating iPSCs from patient-derived cells and differentiating them into specific phenotypes, new personalized models are created for the study and treatment of various pathologies, with the possibility of personalized regenerative approaches with a significantly reduced risk of immune rejection. However, as noted by Scheiner et al. (2013) [253], even autologous iPSC-derived cells can, in some cases, trigger an immune response.

## Figures and Tables

**Figure 1 ijms-26-08255-f001:**
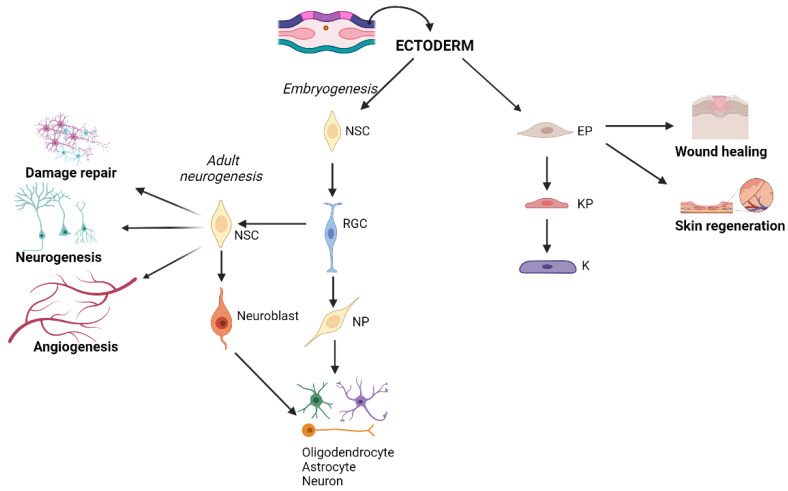
Stem cells from ectoderm-derived tissues with regenerative medicine applications. NSC: neuronal stem cell; RGC: radial glial cells; NP: neuronal progenitor; EP: epithelial progenitor; KP: keratinocyte progenitor; K: keratinocyte. Created in https://BioRender.com.

**Figure 2 ijms-26-08255-f002:**
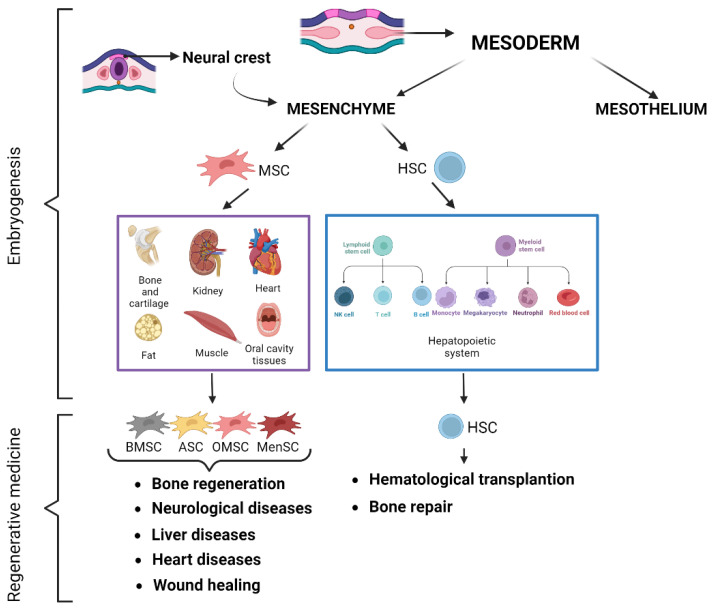
Stem cells from mesoderm-derived tissues and their applications in regenerative medicine. MSC: mesenchymal stem cell; HSC: hematopoietic stem cell. Created in https://BioRender.com.

**Figure 3 ijms-26-08255-f003:**
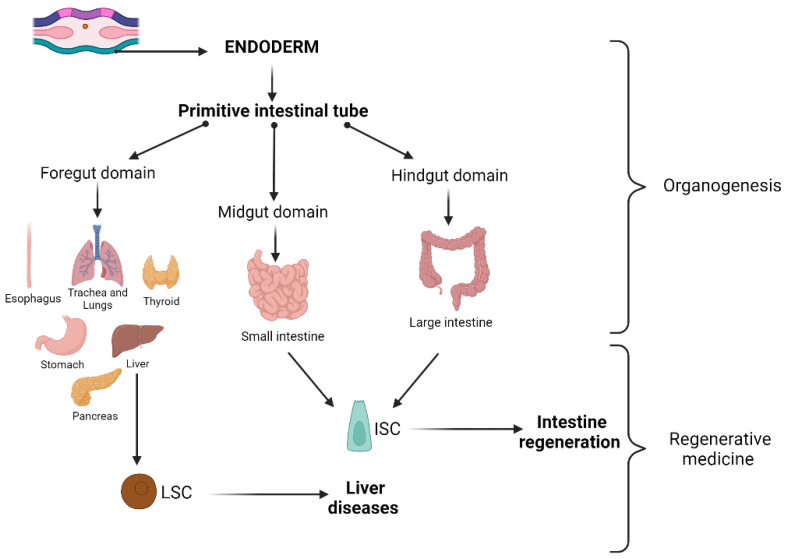
Stem cells from endoderm-derived tissues and their involvement in regenerative medicine. ISC: intestinal stem cell; LSC: liver stem cell. Created in https://BioRender.com.

**Figure 4 ijms-26-08255-f004:**
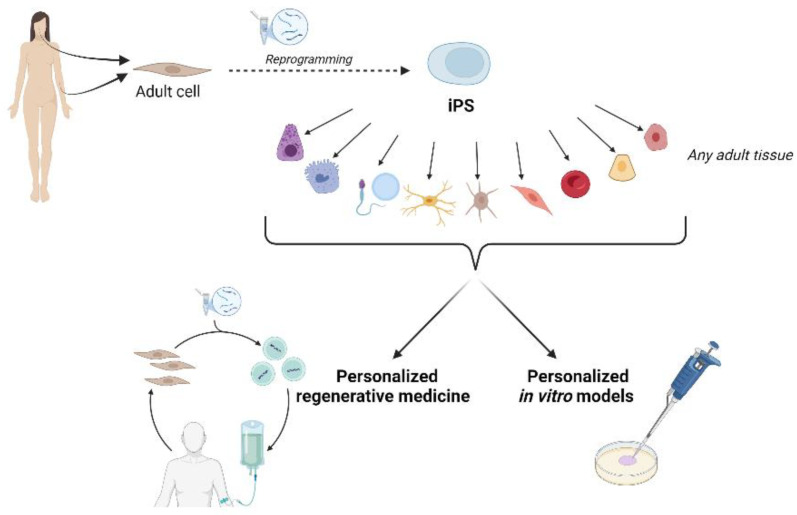
iPSCs (induced pluripotent stem cells) as a new tool for personalized regenerative medicine and for personalized in vitro model constitution. Created in https://BioRender.com.

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
