# Peer review of "Stem Cells in Regenerative Medicine: A Journey from Adult Stem Cells to Induced Pluripotent Cells"

_ijms, 2025, doi:10.3390/ijms26178255_

Round 1
Reviewer 1 Report
Comments and Suggestions for Authors
In the manuscript entitled “Stem Cells in Regenerative Medicine: A Journey from Adult Stem Cells to Induced Pluripotent Cells” the authors explored the crucial role and evolution of stem cells within translational regenerative medicine. Its value lies in offering an extensive and in-depth overview of various stem cell types and their impact.
Overall, the review is robust providing a comprehensive understanding of different stem cell types but also balancing the significant progress made in the field with the persistent challenges that regenerative medicine still needs to overcome.
The manuscript could be improved with some issues to take into account.
The claim that regenerative medicine is "becoming the only therapeutic approach available for some incurable diseases" is too strong. While it offers immense hope, it's rarely the sole option and is often still in experimental stages for many conditions. A more cautious and realistic phrasing, such as "Regenerative medicine presents promising avenues for addressing a wide range of currently incurable diseases [1]," would be more accurate.
Ensure "iPSCs" or "iPS" is used consistently throughout the entire article (title, abstract, body text, keywords).
For Regenerative Medicine: While your current definition is good, the phrase "through different methods and applications [2]" is a bit vague. Consider a more precise phrasing that elaborates on the types of methods or the scope of applications without being overly exhaustive.
For Stem Cells: The definition "Stem cells are cells capable of self-renewal and differentiation into specialized tissue cells" is correct. However, it could be enhanced by integrating the crucial concept of "potency".
Integrate cutting-edge research: this is a significant value to add. The recent finding that iPSCs constitutively express the progesterone receptor (PR) at the protein level (DOI: 10.1007/s12015-024-10776-6) is a crucial piece of information. This detail enhances the fidelity of iPSC-based models for physiological and pathological processes. It demonstrates a more complex cellular aspect linked to reprogramming and makes iPSCs an even more powerful tool for recapitulating molecular events in disease modeling.
Author Response
The claim that regenerative medicine is "becoming the only therapeutic approach available for some incurable diseases" is too strong. While it offers immense hope, it's rarely the sole option and is often still in experimental stages for many conditions. A more cautious and realistic phrasing, such as "Regenerative medicine presents promising avenues for addressing a wide range of currently incurable diseases [1]," would be more accurate.
Thanks to the referee for his comment; we fully agree with his note. The manuscript has now been revised accordingly.
Ensure "iPSCs" or "iPS" is used consistently throughout the entire article (title, abstract, body text, keywords).
Thanks for the correction. The manuscript is now consistent with the term "iPSCs."
For Regenerative Medicine: While your current definition is good, the phrase "through different methods and applications [2]" is a bit vague. Consider a more precise phrasing that elaborates on the types of methods or the scope of applications without being overly exhaustive.
Thanks to the referee for his comment and attention. At that precise point, the sentence is vague because it introduces the subsequent sentence in which the types of methods, etc., are explored in detail as you suggested. Therefore, specifying and describing them in detail in this sentence probably seemed redundant.
For Stem Cells: The definition "Stem cells are cells capable of self-renewal and differentiation into specialized tissue cells" is correct. However, it could be enhanced by integrating the crucial concept of "potency".
Thanks to the referee for his suggestion. We completely agree with his comment. The period has now been improved by adding the suggestions regarding stem cell potency.
Integrate cutting-edge research: this is a significant value to add. The recent finding that iPSCs constitutively express the progesterone receptor (PR) at the protein level (DOI: 10.1007/s12015-024-10776-6) is a crucial piece of information. This detail enhances the fidelity of iPSC-based models for physiological and pathological processes. It demonstrates a more complex cellular aspect linked to reprogramming and makes iPSCs an even more powerful tool for recapitulating molecular events in disease modeling.
We thank the referee for the valuable information on progesterone receptor expression identified in iPSCs and for the interesting article he suggested. The iPSC section has now been updated and improved thanks to this comment.
Reviewer 2 Report
Comments and Suggestions for Authors
The authors review all the different stem cells with their differentiation potentials and applications in therapeutic approaches. The embryonic origine of the stem cell articulate the review. The neuronal stem cell part is very dense in comparison to the other stem cells. However, the review is quite exhaustive and describe well all the adults stem cells used in therapeutic approaches.
Minor points:
-line155: " In progenitor neuronal cells that reside in VZ and SVZ, is not detectable [30]." What is not detectable?
-line 458: " interleuchine" instead of interleukine
- line 460: "BMMSC" what are these cells? are they different from BMSC? and line 462, 500 and 581 same comment
line 492: "ADSCs" what is this acronym for?
-line751: "haematopoietic" What is the curent langage US or UK english? the author should homogenize the used langage
-line812: Diredct instead pf direct
-line 834 :"at epigenetic level of extracellular vesicles miRNAs content" could the authors explain what they mean or rewrite this part of the sentence
Author Response
-line155: " In progenitor neuronal cells that reside in VZ and SVZ, is not detectable [30]." What is not detectable?
Thanks to the referee for his attention and comment. Sox5 is not detectable in progenitor neuronal cells that reside in VZ and SVZ. It was mistakenly not reported; it has now been specified in the text.
-line 458: " interleuchine" instead of interleukine
Thanks. Correction made
- line 460: "BMMSC" what are these cells? are they different from BMSC? and line 462, 500 and 581 same comment
Thanks to the referee for his attention and comment. There was an error; the cells in question are bone marrow mesenchymal stem cells (BMSCs). The errors in the suggested lines have been corrected.
line 492: "ADSCs" what is this acronym for?
Thanks to the referee for his comment. The acronym ASCs refers to adipogenic stem cells in general, different from the acronym ADSCs, which refers to adipose-derived stem cells, i.e., those extracted. The acronym has now been specified in the text. Thank you.
-line751: "haematopoietic" What is the curent langage US or UK english? the author should homogenize the used langage
Thanks to the reviewer for his attention and comments. There was an oversight due to the fact that the concept in reference 193 used the British form "haematopoietic." This has now been corrected to "hematopoietic," homogenizing with the rest of the text.
-line812: Diredct instead pf direct
Thanks. Correction made
-line 834 :"at epigenetic level of extracellular vesicles miRNAs content" could the authors explain what they mean or rewrite this part of the sentence
Thanks to the referee for his request, which allows us to better explain the concept. In the study cited in the line 834, it was demonstrated that reprogramming human gingival mesenchymal stem cells (hGMSCs) using a non-integrating method not only there was a change in gene and protein expression through the expression of pluripotency markers but also a rearrangement of membrane cluster differentiation (CD) compared to the parental cell line. Furthermore, an epigenetic change was also highlighted in the reprogrammed cell line, since the extracellular vesicles produced by the iPSCs, contained miRNA content with different expression compared to the miRNA content of the vesicles produced by the parental line, with eight miRNAs appearing de novo in the iPSC vesicles that were not present in the parental ones. The concept has now been further clarified in the text. Thank you.
Reviewer 3 Report
Comments and Suggestions for Authors
The manuscript “Stem cells in regenerative medicine: a journey from adult stem cells to induced pluripotent cells” by Rocca et. al summarizes different types of stem cells, their differentiation potential, and their applications in regenerative therapy. This includes neural stem cells, stem cells from mesodermal-derived tissues, and endodermal-derived tissues. The paper also details ESCs and iPSCs, acknowledging the debates and their promising applications.
Line 17-21. iPSCs are not derived from embryonic germ layers. Please rewrite this sentence for better clarity.
Line 43: “Regenerative medicine offers cellular therapeutic perspectives for various diseases” is this a heading for the below paragraph?
Line 99. Check grammar “led approved therapies”
Line 96: To replace “origin tissue” with tissue of origin.
Line 97: tissue damaged cells?
Line 121-123: Re-structure the sentence, not clear.
Line 155: Unclear sentence “In progenitor neuronal cells that reside in VZ and SVZ, is not detectable”
Line 171-174: Long, unclear sentence
Line 178: to replace “polysialated” with polysialylated
Line 192-195: Unclear, please rephrase this sentence “CD71 could be used to distinguish more mature neuronal derivatives, while early committed NSCs express CD133, CD15. A2B5, CD44 and CD140a are markers of glial, astrocytic and oligodendrocytic precursors, respectively. Differentiated neuronal cells express CD56 and CD24 .
Line 218: not “an human”
Line 281: The skin is an immunocompetent organ, therefore the only way to permanently repair severe skin wounds is to use autologous skin grafts. This is a scientifically incorrect and is immediately contradicted by the authors' own content.
Line 273-276: Unclear sentence. Reframe.
Line 371: “At 2009” ?
Line 395: Reframe the sentence “One of the main limitations is that the BMSCs capacities decrease with aging”
Line 404: Are ADSCs and the ASCs same?
Line 420: “an high”
Line 445: Incomplete sentence “While the anti-apoptotic effect is mediated by the secretion mainly of insulin- 445 like growth factor-1 (IGF-1)”
Line 513: HMC-II to MHC
Line 549: Gingiva is more commonly used not gengiva
Line 611: Mis-spelled “diabete”
Line 747: Misspelled albumin
Line 767-768: Rephrase the sentence “Furthermore, in mice models with carbon tetrachloride (CCl4)-induced fibrosis and meanwhile transplanted weekly with human LSCs, it was seen that advanced fibrosis (bridging fibrosis and cirrhosis) was prevented by LSCs transplantation”
Line 812: “Diredct” spellcheck please
Line 848: SKP: not defined prior to its first use.
Line 892: Even autologous iPSC-derived cells can sometimes elicit an immune response. Refer to this paper and rephrase accordingly. Scheiner et al 2013. The Potential for Immunogenicity of Autologous Induced Pluripotent Stem Cell-derived Therapies.
Comments on the Quality of English LanguageAlthough the content is relevant, the article is poorly drafted without proper editing, proofreading or spellcheck.
Author Response
Line 17-21. iPSCs are not derived from embryonic germ layers. Please rewrite this sentence for better clarity.
Thanks to the referee for the comment. The referenced lines do not report that iPSCs derive from germ layers, but rather that the review would have summarized the most studied and recently discovered types of stem cells, from adult stem cells to the category of innovative induced pluripotent stem cells. Furthermore, the purpose of the review, stated in the following sentences, also specifies that the stem cells involved in regenerative medicine approaches would be analyzed using a classification based on embryonic origin, specifically reporting the individual categories (stem cells from ectodermal-derived tissues, stem cells from mesodermal-derived tissues, stem cells from endodermal-derived tissues, and iPSCs), in which iPSCs are considered as a separate category and as stem cells derived from germ layers.
Line 43: “Regenerative medicine offers cellular therapeutic perspectives for various diseases” is this a heading for the below paragraph?
We thank the referee for his comments and careful review of our manuscript. No, "Regenerative medicine offers cellular therapeutic perspectives for various diseases" is not a paragraph title but a sentence in the introduction to introduce the concepts presented later.
Line 99. Check grammar “led approved therapies”
Thanks to the referee for pointing out the grammatical error. It has now been corrected.
Line 96: To replace “origin tissue” with tissue of origin.
Thanks to the referee for pointing out the error. It has been now corrected.
Line 97: tissue damaged cells?
Thanks to the referee for his attention. There was a typo. It has now been corrected to "tissue cells damaged".
Line 121-123: Re-structure the sentence, not clear.
Thanks to the referee for his suggestion, which allowed us to improve the manuscript. The sentence has now been rewritten more clearly
Line 155: Unclear sentence “In progenitor neuronal cells that reside in VZ and SVZ, is not detectable”
Thanks to the referee for the suggestion. The sentence has now been corrected.
Line 171-174: Long, unclear sentence
Thanks to the referee for his suggestion. The sentence has now been divided into two separate sentences and rewritten clearly.
Line 178: to replace “polysialated” with polysialylated
Thanks. Correction made.
Line 192-195: Unclear, please rephrase this sentence “CD71 could be used to distinguish more mature neuronal derivatives, while early committed NSCs express CD133, CD15. A2B5, CD44 and CD140a are markers of glial, astrocytic and oligodendrocytic precursors, respectively. Differentiated neuronal cells express CD56 and CD24 .
Thanks to the referee. The period has now improved.
Line 218: not “an human”
Thanks. Correction made.
Line 281: The skin is an immunocompetent organ, therefore the only way to permanently repair severe skin wounds is to use autologous skin grafts. This is a scientifically incorrect and is immediately contradicted by the authors' own content.
Thanks to the referee for carefully reviewing our manuscript and to allow a scientific comparison. We didn't understand well what the reviewer was referring when he said " This is a scientifically incorrect". If he was referring to the fact that autologous skin grafts are the only solution for repairing severe skin wounds, given that the skin is an immunocompetent organ, we specified in the text that TO DATE, this is the clinical solution available. If, however, he was referring to the fact that the concept of the skin being an immunocompetent organ may be incorrect, we report the concept reported that is explicitly published in the paper cited in reference 74 (Yang, R.; Yang, S.; Zhao, J.; Hu, X.; Chen, X.; Wang, J.; Xie, J.; Xiong, K. Correction to: Progress in studies of epidermal stem cells and their application in skin tissue engineering. Stem cell research & therapy 2022, 13, 183, doi:10.1186/s13287-022-02868-2.), where the exact words in the introduction are: "As the skin is an immunocompetent organ, autologous skin grafts are the only feasible method to cover serious skin wounds permanently." There is considerable scientific evidence on the skin immunocompetence. Just to name a few:
Guglielmi G. The skin's 'surprise' power: it has its very own immune system. Nature. 2025 Jan;637(8045):260. doi: 10.1038/d41586-024-04068-9. PMID: 39672931.
Zhang, C., Merana, G.R., Harris-Tryon, T. et al. Skin immunity: dissecting the complex biology of our body’s outer barrier. Mucosal Immunol 15, 551–561 (2022). https://doi.org/10.1038/s41385-022-00505-y
Kabashima, K., Honda, T., Ginhoux, F. et al. The immunological anatomy of the skin. Nat Rev Immunol 19, 19–30 (2019). https://doi.org/10.1038/s41577-018-0084-5
Line 273-276: Unclear sentence. Reframe.
Thanks to the referee for allowing us to improve the manuscript with his suggestions. The sentence has now been rewritten more clearly and precisely.
Line 371: “At 2009” ?
Thanks to the referee for his attention. We apologize for the typo; it has now been corrected to “In 2009”.
Line 395: Reframe the sentence “One of the main limitations is that the BMSCs capacities decrease with aging”
Thanks to the referee for his suggestion. The sentence has now been rephrased.
Line 404: Are ADSCs and the ASCs same?
Thanks to the referee for his question.No, ADSCs and ASCs are not exactly the same thin. The acronym ASCs refers to adipogenic stem cells in general, different from the acronym ADSCs, which refers to adipose-derived stem cells, i.e., those extracted. The acronym has now been specified in the text. Thank you.
Line 420: “an high”
Thanks. Correction made.
Line 445: Incomplete sentence “While the anti-apoptotic effect is mediated by the secretion mainly of insulin- 445 like growth factor-1 (IGF-1)”
hanks to the referee for his correction. The sentence has now been updated.
Line 513: HMC-II to MHC
Thanks. Correction made.
Line 549: Gingiva is more commonly used not gengiva
Thanks. Correction made.
Line 611: Mis-spelled “diabete”
Thanks to the referee for his attention. We apologize for the typo; it has now been corrected
Line 747: Misspelled albumin
Thanks to the referee for his attention. We apologize for the typo; it has now been corrected
Line 767-768: Rephrase the sentence “Furthermore, in mice models with carbon tetrachloride (CCl4)-induced fibrosis and meanwhile transplanted weekly with human LSCs, it was seen that advanced fibrosis (bridging fibrosis and cirrhosis) was prevented by LSCs transplantation”
Thanks to the referee for his suggestion. The sentence has now been rephrased.
Line 812: “Diredct” spellcheck please
Thanks. Correction made.
Line 848: SKP: not defined prior to its first use.
Thanks to the referee for his attention. SKP: Skin-derived precursor cells. it has now been reported in the text.
Line 892: Even autologous iPSC-derived cells can sometimes elicit an immune response. Refer to this paper and rephrase accordingly. Scheiner et al 2013. The Potential for Immunogenicity of Autologous Induced Pluripotent Stem Cell-derived Therapies.
Thanks to the referee for the interesting suggested paper. The period has been updated in accordance with the referee's comments.